# SynBench: Task-Agnostic Benchmarking of Pretrained Representations using Synthetic Data

**Ching-Yun Ko**[*]
MIT
cyko@mit.edu

**Pin-Yu Chen**
IBM Research
pin-yu.chen@ibm.com

**Jeet Mohapatra**
MIT
jeetmo@mit.edu

**Payel Das**
IBM Research
daspa@us.ibm.com

**Luca Daniel**
MIT
luca@mit.edu

## Abstract

Recent success in fine-tuning large models, that are pretrained on broad data at scale, on downstream tasks has led to a significant paradigm shift in deep learning, from task-centric model design to task-agnostic representation learning and task-specific fine-tuning. As the representations of pretrained models are used as a foundation for different downstream tasks, this paper proposes a new task-agnostic framework, *SynBench*, to measure the quality of pretrained representations using synthetic data. Our framework applies to a wide range of pretrained models taking continuous data inputs and is independent of the downstream tasks and datasets. Evaluated with several pretrained vision transformer models, the experimental results show that our SynBench score well matches the actual linear probing performance of the pre-trained model , and can inform the design of robust linear probing on pretrained representations to mitigate the robustness-accuracy tradeoff in downstream tasks. Code are available at `https://github.com/IBM/synbench`.

## 1 Introduction

In recent years, the use of large pretrained neural networks for efficient fine-tuning on downstream tasks has prevailed in many application domains such as vision, language, and speech. Instead of designing task-dependent neural network architectures for different downstream tasks, the current methodology focuses on the principle of task-agnostic pretraining and task-specific finetuning, which uses a neural network pretrained on a large-scale dataset (often in a self-supervised or unsupervised manner) to extract generic representations of the input data, which we call *pretrained representations* for simplicity. The pretrained representations are then used as a foundation [1] to solve downstream tasks by training a linear head (i.e., linear probing) on the data representations with the labels provided by a downstream dataset, or by simply employing zero-shot inference.

As large pretrained models are shown to achieve state-of-the-art performance on a variety of downstream tasks with minimal fine-tuning, there is an intensified demand for using pretrained represen-

---

[*]Work done as a summer intern at IBM Research.

NeurIPS 2022 Workshop on Synthetic Data for Empowering ML Research.

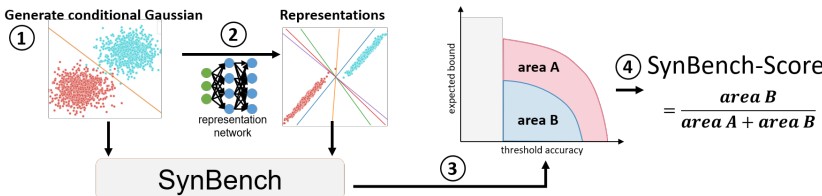

Figure 1: Overview of our SynBench framework. Step 1: generate class conditional Gaussian and form the inputs to the pretrained model; Step 2: gather rendered representations; Step 3: measure and plot the expected bound under a range of threshold accuracy for both input raw data and representations according to equation 2; Step 4: calculate SynBench score by the relative area under curve of the representations to the input data in the expected bound-threshold accuracy plot.

tations from a large model for efficient finetuning. However, if the underlying pretrained model is at risk, such as lacking robustness to adversarial examples, this trending practice of pretraining and fine-tuning also signifies the immediate damage to all downstream tasks. To address this emerging challenge, we propose a novel framework named *SynBench* to evaluate the quality of pretrained representations, in terms of quantifying the tradeoff between standard accuracy and adversarial robustness to input perturbations. Specifically, SynBench uses synthetic data generated from a conditional Gaussian distribution to establish a reference characterizing the robustness-accuracy tradeoff based on the Bayes optimal linear classifiers. Then, SynBench obtains the representations of the same synthetic data from the pretrained model and compare them to the reference for performance benchmarking. Finally, we define the ratio of area-under-curves in robustness-accuracy characterization as a quantifiable metric of the quality of pretrained representations. The entire procedure of SynBench is illustrated in Figure 1. Our SynBench framework features the following key advantages.

1. *Soundness*: We formalize the fundamental tradeoff in robustness and accuracy of the considered conditional Gaussian model and use this characterization as a reference to benchmark the quality of pretrained representations.

2. *Task-independence*: Since the pretraining of large models is independent of the downstream datasets and tasks (e.g., through self-supervised or unsupervised training on broad data at scale), the use of synthetic data in SynBench provides a task-agnostic approach to evaluating pretrained representations without the knowledge of downstream tasks and datasets.

3. *Completeness and privacy*: The flexibility of generating synthetic data (e.g., by adopting a different data sampling procedure) offers a good proxy towards a more comprehensive evaluation of pretrained representations when fine-tuned on different downstream datasets, especially in the scenario when the available datasets are not representative of the entire downstream datasets. Moreover, the use of synthetic data enables full control and simulation over data size and distribution, protects data privacy, and can facilitate model auditing and governance.

## 2 SynBench: Methodology and Evaluation

On the whole, we want to measure the idealized robustness-accuracy tradeoff using synthetic data. By propagating the realizations through representation networks, we can also measure the robustness-accuracy tradeoff for representations. We start the section by giving the desired synthetic data.

### 2.1 Linear Classifier

We consider binary classification problems with data pair $(x, y)$ generated from the mixture of two Gaussian distributions

$$x|y \sim \mathcal{N}(y\mu, \Sigma), \tag{1}$$

where $y \in \mathcal{C} = \{+1, -1\}$, $P(y = +1) = p$, and $P(y = -1) = 1 - p$. When sampling from this idealized distribution, we eliminate the factor of data bias and can benchmark the robustness degradation in an ideal setting.

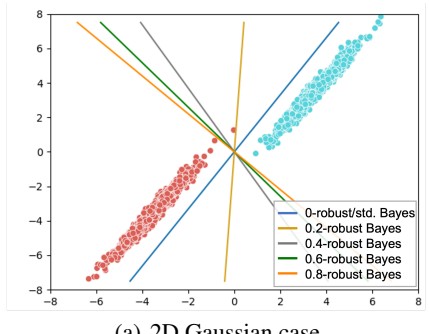

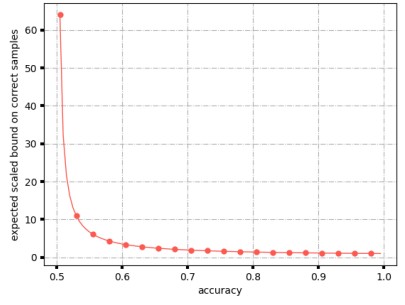

| (a) 2D Gaussian case | (b) Theoretical robustness-accuracy tradeoff |

Figure 2: Illustration of robustness-accuracy tradeoff suggested by $\epsilon$-robust Bayes optimal classifiers. Figure (a) depicts a class conditional 2D Gaussian case with decision boundaries drawn by $\epsilon$-robust Bayes optimal classifiers of varying $\epsilon$ values. Figure (b) draws the theoretically characterized robustness-accuracy tradeoff given in Result 2.3.

For a given classifier $f$ and input $x$ with $f(x) = y$, where $y$ is the predicted label, it is not rational for the classifier to respond differently to $x + \delta$ than to $x$ for a small perturbation level measured by $\|\delta\|_p$, i.e. inconsistent top-1 prediction [2, 3]. Therefore, the level of (adversarial) robustness for a classifier can be measured by the minimum magnitude of perturbation that causes misclassification, i.e. $\min_{\delta:f(x+\delta)\neq f(x)} \|\delta\|_p$. For a generic function $f$, solving the optimization problem exactly is hard [4, 5]. Luckily, one can readily solve for the optimization if $f$ is affine [6].

In the following, we will exploit this point and consider the linear classifier that minimizes the robust classification error. An ideal candidate classifier for the class conditional Gaussian (equation 1) is specified by the robust Bayes optimal classifier [7, 8]. Specifically, it is stated that the optimal robust classifier (with a robust margin $\epsilon$) for data generated from equation 1 is a linear classifier $f(x) = sign(w_0^T x)$, where $w_0 = \Sigma^{-1}(\mu - z_\Sigma(\mu))$, $z_\Sigma(\mu) = \arg\min_{\|z\|_2 \leq \epsilon}(\mu - z)^T \Sigma^{-1}(\mu - z)$, and $sign(\cdot)$ is the typical sign function. To simplify the exposition, we focus on $\ell_p$ with $p = 2$ in the remainder of this paper (We put Bayes optimal $\ell_\infty$ robust classifier in the appendix **??**). We derive the following result as a direct application of the fact:

**Result 2.1.** *For samples $x$ following the conditional Gaussian in equation 1 with $\Sigma = I_d$ ($d$ by $d$ identity matrix), given an $\ell_2$ adversarial budget $\epsilon \leq \|\mu\|_2$, the robust Bayes optimal classifier has the decision margin $\delta$ lower bounded by $\frac{|q/2 - x^T\mu(1-\epsilon/\|\mu\|_2)|}{(1-\epsilon/\|\mu\|_2)\|\mu\|_2}$, where $q = ln\{(1-p)/p\}$. With $p = \frac{1}{2}$, the lower bounds become $\frac{|x^T\mu|}{\|\mu\|_2}$.*

Since the bound is subject to the distance between two Gaussians, we scale the bound by $\|\mu\|$ and obtain the minimal scaled perturbation $\bar{\delta}$ as $\|\bar{\delta}\|_2 \geq \frac{|x^T\mu|}{\|\mu\|_2^2}$. We note that when the classes are balanced, for samples generated from 1 and $\Sigma = \sigma^2 I_d$, all $\epsilon$-robust Bayes optimal classifier overlap with each other. However, for data generated from conditional Gaussian with general covariance matrices, the $\epsilon$ of a $\epsilon$-robust Bayes classifier specifies the desired size of margin and demonstrates the robustness accuracy tradeoff (see Figure 2). We give an illustrative 2D class conditional Gaussian example in Figure 2(a), where as the $\epsilon$ increases, robust Bayes optimal classifier rotates counterclockwise, leading to misclassifications, but overall bigger margins.

## 2.2 Objective

For a given representation network, we are interested in evaluating the expected bounds under a thresholding accuracy $a_t$, i.e. $\mathbb{E}_{\mu\sim\mathbb{P}_\mu, \Sigma\sim\mathbb{P}_\Sigma, x-\bar{\mu}|y\sim\mathcal{N}(y\mu,\Sigma)} \left[\|\bar{\delta}\|_2 \mid \hat{y}^*(x) = y, a > a_t\right]$, where $\mathbb{P}_\mu$ and $\mathbb{P}_\Sigma$ characterize the probability density function of the synthetic data manifold of interest, and $\bar{\mu}$ is a translation vector allowing non-symmetric class conditional Gaussian. Here, without the prior of applications, we assume $\mu = s \cdot 1_d/\sqrt{d}$, where $s$ denotes a random variable that follows uniform distribution and $1_d/\sqrt{d}$ is the normalized all-ones vector. For simplicity, we let $\Sigma = I_d$. Formally,

we define $E_{\theta,\epsilon}(a_t)$ as

$$E_{\theta,\epsilon}(a_t) = \mathbb{E}_{s,x}\left[\|\bar{\delta}\|_2 \mid \hat{y}^*(x) = y, a > a_t, \mu = s \cdot 1_d/\sqrt{d}, \Sigma = I_d\right]$$

$$= \frac{1}{n}\sum_i \mathbb{E}_{x-\bar{\mu}|y\sim\mathcal{N}(ys_i\cdot 1_d/\sqrt{d}, I_d)}\left[\|\bar{\delta}\|_2 \mid \hat{y}^*(x) = y\right] \mathbb{1}_{a(s_i\cdot 1_d/\sqrt{d}, I_d, \epsilon) > a_t}, \quad (2)$$

where $\mathbb{1}_{a(s_i\cdot 1_d/\sqrt{d}, I_d, \epsilon) > a_t}$ is the indicator function specifying the $s_i, \epsilon$-dependent $a$ that surpasses the threshold accuracy $a_t$. In the following sections, we will illustrate how to calculate the inner expectation term $\mathbb{E}_{x-\bar{\mu}|y\sim\mathcal{N}(ys_i\cdot 1_d/\sqrt{d}, I_d)}\left[\|\bar{\delta}\|_2 \mid \hat{y}^*(x) = y\right]$ for both the raw data and representations.

### 2.2.1 Raw data

Denote the CDF of the standard normal distribution as $\Phi$, we rely on the below two theorems to obtain the expected bounds for the generated synthetic data:

**Result 2.2.** *Assume a balanced dataset ($p = \frac{1}{2}, q = 0$) where samples $x$ follow the general conditional Gaussian $x|y = 1 \sim \mathcal{N}(\mu_1, \Sigma), x|y = -1 \sim \mathcal{N}(\mu_2, \Sigma)$, given an $\ell_2$ adversarial budget $\epsilon$, the robust Bayes optimal classifier gives a standard accuracy of $\Phi\left(\frac{\tilde{\mu}^T\Lambda^{-1}(\tilde{\mu}-z_\Lambda(\tilde{\mu}))}{\|\Lambda^{-1}(\tilde{\mu}-z_\Lambda(\tilde{\mu}))\|_\Lambda}\right)$, where $\tilde{\mu} = F^T\frac{\mu_1-\mu_2}{2}$, $F\Lambda F^T = \Sigma$ is the economy-size (thin) decomposition with nonzero eigenvalues, and $z_\Lambda$ is the solution of the convex problem $\arg\min_{\|z\|_2\leq\epsilon}(\tilde{\mu}-z)^T\Lambda^{-1}(\tilde{\mu}-z)$.*

While Result 2.2 gives us the theoretical classification accuracy as a function of synthetic conditional Gaussian parameters, the following result establishes a direct link between the expected scaled bound and accuracy (i.e. robustness-accuracy tradeoff).

**Result 2.3.** *Assume a balanced dataset ($p = \frac{1}{2}, q = 0$) where samples $x$ follow the general conditional Gaussian $x|y = 1 \sim \mathcal{N}(\mu_1, \Sigma), x|y = -1 \sim \mathcal{N}(\mu_2, \Sigma)$, given an $\ell_2$ adversarial budget $\epsilon$, the robust Bayes optimal classifier gives an expected scaled bound of $\mathbb{E}\left[\|\bar{\delta}_x\|_2 \mid \hat{y}^*(x) = y\right] \geq \frac{1}{\sqrt{2\pi}}\frac{1}{a\Phi^{-1}(a)}e^{-\frac{1}{2}\left(\Phi^{-1}(a)\right)^2} + 1$, where $a$ denotes the standard accuracy.*

The subscript $x$ in the expected scaled bound $\mathbb{E}\left[\|\bar{\delta}_x\|_2 \mid \hat{y}^*(x) = y\right]$ indicates the raw data space, to distinguish from the scaled bound to be derived for representations. We highlight that Result 2.3 directly gives a robustness-accuracy tradeoff. We plot the expected scaled bound as a function of accuracy in Figure 2(b). This tradeoff holds true when the data follow the conditional Gaussian exactly. In the proposed SynBench framework, we treat this theoretically-derived robustness-accuracy tradeoff as the reference, enabling a fair comparison among representations induced by different pretrained models.

By now, with Result 2.3, we can already calculate the inner expectation term in equation 2 for the raw data and provide a theoretically-sounded characterization of robustness-accuracy tradeoff of Bayes optimal classifiers on raw data.

### 2.2.2 Representations

Given a pretrained network , we gather the representations of the Gaussian realizations and quantify the desired bound induced by robust Bayes optimal classifier in the representation space. When deriving the robust Bayes optimal classifier, we model the representations by a general conditional Gaussian $z|y = 1 \sim \mathcal{N}(\mu_1, \Sigma), z|y = -1 \sim \mathcal{N}(\mu_2, \Sigma)$. It is worthwhile to note that now the Bayes optimal classifier does not necessarily coincide with robust Bayes optimal classifier even when the dataset is class balanced (see Figure 2(a)). The following result is essential to the development of the robustness-accuracy quantification of representations.

**Result 2.4.** *For representations $z$ following the general class-balanced conditional Gaussian $z|y = 1 \sim \mathcal{N}(\mu_1, \Sigma)$, $z|y = -1 \sim \mathcal{N}(\mu_2, \Sigma)$, given an $\ell_2$ adversarial budget $\epsilon$, the robust Bayes optimal classifier has the decision margin $\delta_z$ lower bounded by $\frac{|(z-\frac{\mu_1+\mu_2}{2})^T F\Lambda^{-1}(\tilde{\mu}-z_\Lambda(\tilde{\mu}))|}{\|F\Lambda^{-1}(\tilde{\mu}-z_\Lambda(\tilde{\mu}))\|_2}$, and a scaled bound of $\|\bar{\delta}_z\|_2 \geq \frac{|(z-\frac{\mu_1+\mu_2}{2})^T F\Lambda^{-1}(\tilde{\mu}-z_\Lambda(\tilde{\mu}))|}{|\tilde{\mu}^T\Lambda^{-1}(\tilde{\mu}-z_\Lambda(\tilde{\mu}))|}$, where $\tilde{\mu} = F^T\frac{\mu_1-\mu_2}{2}$, $F\Lambda F^T = \Sigma$ is the economy-size (thin) decomposition with nonzero eigenvalues, and $z_\Lambda$ is the solution of the convex problem $\arg\min_{\|z\|_2\leq\epsilon}(\tilde{\mu}-z)^T\Lambda^{-1}(\tilde{\mu}-z)$.*

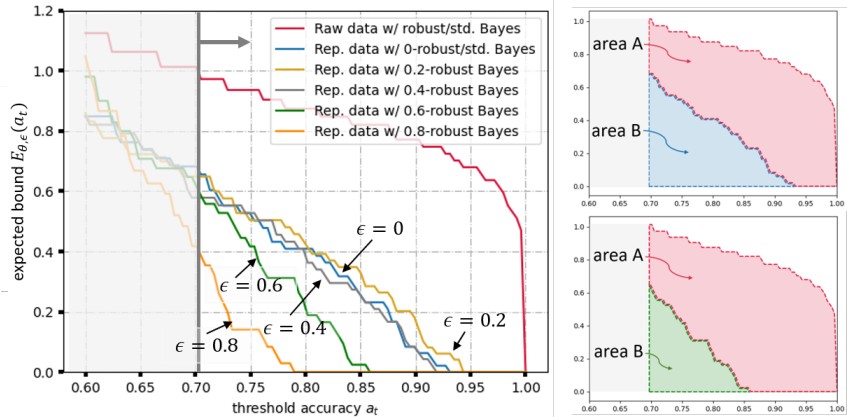

Figure 3: An example of the robustness-accuracy quantification for ViT-B/16 representations. (Left) The expected bound as a function of the threshold accuracy, $E(a_t)$ and $E_{\theta,\epsilon}(a_t)$, for $\epsilon = 0 \sim 0.8$. (Right) The SynBench-Score = area B/(area A + area B) (equation 3) for $\epsilon = 0$ (top) and $0.6$ (bottom).

### 2.3 Robustness-Accuracy Quantification of Representations

Recall that we want to calculate

$$E_{\theta,\epsilon}(a_t) = \frac{1}{n}\sum_i \mathbb{E}_{x|y \sim \mathcal{N}(ys_i \cdot 1_d/\sqrt{d}, I_d)} \left[ \|\bar{\delta}\|_2 \mid \hat{y}^*(x) = y \right] \mathbb{1}_{a(s_i \cdot 1_d/\sqrt{d}, I_d, \epsilon) > a_t}$$

for both raw data and the representations (i.e. $\|\bar{\delta}_x\|$ and $\|\bar{\delta}_z\|$). We treat the expected bounds of the raw data under a threshold accuracy as the reference. Given a representation network, we compare the expected bounds of the representations rendered by representation networks with the reference.

We take $s \sim \mathcal{U}\{0.1, 5\}$ under the guidance of Result 2.2. Specifically, as Results 2.2 gives an analytical expected accuracy for class conditional Gaussian, we can obtain the desired range of $s$ by giving the accuracy. Now since we are interested in having the reference as a class conditional Gaussian that yields accuracy from 55% to almost 100%, we set the starting and ending $s$ by the fact that $\Phi(0.1) \sim 0.55$ and $\Phi(5) \sim 1.0$. We reiterate that with more accurate modelling of the data manifold of interest, SynBench can give more precise capture of the pretrained representation performance.

When the data is perfect Gaussian (e.g. raw data), we calculate $E_{\theta_{\mathrm{raw}},\epsilon}(a_t)$ with the help of Result 2.3. Since $E_{\theta_{\mathrm{raw}},\epsilon}(a_t)$ of the raw data is defined for a specific representation network parameter $\theta_{\mathrm{raw}}$, and all the $\epsilon$-robust classifiers overlap with each other, we further denote it by $E(a_t)$ to differentiate it from that of the representations. For representations, we calculate $E_{\theta,\epsilon}(a_t)$ with the help of Result 2.4 and the expectation is estimated empirically. We show an example of the probing results in Figure 3.

To integrate over all the desired threshold accuracy, we use the area under the curve (AUC) and give the ratio to the reference by

$$\text{SynBench-Score}(\theta, \epsilon, a_t) = \frac{\int_{a_t}^1 E_{\theta,\epsilon}(a)da}{\int_{a_t}^1 E(a)da}, \tag{3}$$

which correspond to area B/(area A + area B) in Figure 3. Larger value of SynBench-Score implies better probing performance on pretrained representations.

## 3 Experimental Results

In this experiment, we exemplify the use of SynBench given a pretrained representation network. In order to compare among network attributes, it is desirable to control the variates. In the appendix Table 4, we list severeal pretrained vision transformers (ViTs)[9, 10] from *Pytorch Image Models* package and make comparisons to our best knowledge. We note that the performance of these models might be nuanced by scheduler, curriculum, and training episodes, which are not captured in the above table. To provide a comprehensive evaluation, we give SynBench-Score$(\theta, \epsilon, a_t)$ with $a_t$ ranging

| $a_t = 0.7$ | $\epsilon = 0$ | $\epsilon = 0.2$ | $\epsilon = 0.4$ | $\epsilon = 0.6$ | $\epsilon = 0.8$ | CIFAR10 | CIFAR10-c |
|---|---|---|---|---|---|---|---|
| ViT-B/16 | 0.33 | 0.37 | 0.32 | 0.20 | 0.06 | 95.0 | 81.2 |
| ViT-B/16-in21k | 0.20 | 0.23 | 0.18 | 0.07 | 0.01 | 89.6 | 71.4 |

Table 1: Comparisons on the finetuning procedure in pretraining. We report the SynBench-Score of ViTs with or without finetuning pretrained representations, and the standard linear probing accuracy on CIFAR10 and transfer accuracy on CIFAR10-c.

| $a_t$ | Model | $\epsilon = 0$ | $\epsilon = 0.2$ | $\epsilon = 0.4$ | $\epsilon = 0.6$ | $\epsilon = 0.8$ | CIFAR10 | CIFAR10-c |
|---|---|---|---|---|---|---|---|---|
|     | ViT-Ti/16 | 0.01 | 0 | 0 | 0 | 0 | 81.9 | 59.1 |
| 0.7 | ViT-B/16 | 0.33 | 0.37 | 0.32 | 0.20 | 0.06 | 95.0 | 81.2 |
|     | ViT-L/16 | 0.26 | 0.33 | 0.30 | 0.22 | 0.11 | 98.0 | 90.3 |
|     | ViT-Ti/16 | 0 | 0 | 0 | 0 | 0 | 81.9 | 59.1 |
| 0.8 | ViT-B/16 | 0.19 | 0.23 | 0.17 | 0.04 | 0 | 95.0 | 81.2 |
|     | ViT-L/16 | 0.12 | 0.21 | 0.18 | 0.09 | 0 | 98.0 | 90.3 |
|     | ViT-Ti/16 | 0 | 0 | 0 | 0 | 0 | 81.9 | 59.1 |
| 0.9 | ViT-B/16 | 0.02 | 0.04 | 0.01 | 0 | 0 | 95.0 | 81.2 |
|     | ViT-L/16 | 0 | 0.04 | 0.03 | 0 | 0 | 98.0 | 90.3 |

Table 2: Comparisons on the model sizes. The SynBench-Score of ViTs of different sizes, and the standard linear probing accuracy on CIFAR10 and transfer accuracy on CIFAR10-c.

from 0.7 to 0.9, and $\epsilon$ from 0 to 0.8. Due to space limit, some $a_t$ results are deferred to the appendix. The runtime of SynBench depends on the number of outcomes of the discrete uniform distribution $\mathcal{U}\{0.1, 5\}$. For one $s \sim \mathcal{U}\{0.1, 5\}$, it costs 59 seconds to generate 2048 Gaussian samples, 37 and 81 seconds to obtain the SynBench-Score for ViT-B/16 and ViT-L/16 on one GeForce RTX 2080 super.

Apart from the task-agnostic metrics SynBench-Score developed in this paper, we also report linear probing accuracy on CIFAR10 and CIFAR10-c [11] to validate the standard and transfer accuracy (use the probing layer trained on CIFAR10 to probe CIFAR10-c). We emphasize that SynBench-Score offers a quantifiable score for robustness-accuracy performance benchmarking and is intrinsically a task-agnostic evaluation of the pretrained model. Therefore, although SynBench-Score may share trends with empirical real-life tasks (e.g. CIFAR10), it is meant to characterize a general behavior of the pretrained representations.

**Fine-tuned pretraining representation.** When applying a pretrained representation network to the desired task, one can either only train a linear head on top of a fixed pretrained model, or perform fine-tuning of both the representation network and the linear head. Thus, in Table 1, we investigate how the fine-tuning process is affecting the representation networks. Specifically, both networks in Table 1 is pretrained on Imagenet 21k with supervision. After the pretraining, ViT-B/16 is further finetuned on Imagenet 1k. Interestingly, SynBench-Score shows that this finetuning is beneficial as improvements are witnessed across all $\epsilon$ with SynBench-Score, which well match the empirical observation give by CIFAR10 and CIFAR10-c and prior results [12].

**Model size.** In Table 2, we compare ViTs of different sizes. Specifically, we perform SynBench on ViT-Tiny, ViT-Base, and ViT-Large with patch size being 16. The model parameter $\theta$ is provided by the pretrained model. It is noticeable that ViT-B/16 is generally on par with ViT-L/16. When we set the threshold accuracy to be higher values, ViT-L/16 starts to give slightly better evaluations especially with larger $\epsilon$. One interesting observation is that for each model, SynBench score is not necessarily monotonic in $\epsilon$, which indicates standard linear probing (i.e., $\epsilon = 0$)

| | $\arg\max_\epsilon$ SynBench-Score | $\Delta$ robust linear probing mean (CIFAR10/-c) |
|---|---|---|
| ViT-Ti/16 | 0 | 69.7**+0** |
| ViT-B/16 | 0.2 | 87.6**+1.0** |
| ViT-L/16 | 0.2 | 94.2**+0.4** |
| ViT-B/16-in21k | 0.2 | 80.0**+0.4** |
| ViT-B/32 | 0.2 | 83.9**+0.8** |

Table 3: CIFAR10 and CIFAR10-c accuracy changes using $\epsilon$-robust linear probing with $\epsilon = \arg\max_\epsilon$ SynBench-Score.

may not be the most effective way to probing pretrained representations in terms of robustness-accuracy performance, which is consistent with recent findings [13]. See the "Robust linear probing" paragraph below for detailed analysis. We also observe that larger models exhibit better resilience (slower reduction in SynBench score) as $\epsilon$ increases.

**Robust linear probing.** According to Table 2, 0.2-robust Bayes classifiers consistently give better scores compared to 0-robust (standard) Bayes classifiers with ViT-B/16 and ViT-L/16. This offers us a quick way of gauging the suitable downstream robust probing parameter for the given pretrained model. We stipulate that observing a 0.2-robust Bayes classifier to yield better SynBench-Score than a 0-robust Bayes classifier may suggest the pretrained network to produce representations that have better overall performance with linear classifiers trained by 0.2-robust linear probing. We validate this by performing robust linear probing on representations rendered by ViTs for CIFAR10 classifications. From Table 3, we see that robust linear probing with $\epsilon = \arg\max_\epsilon$ SynBench-Score generally gives a decent robustness-accuracy tradeoff. We refer the readers to the appendix for the robust linear probing procedures and a more complete table on $\epsilon$-robust linear probing results with different $\epsilon$.

## 4 Discussion and Conclusion

In this paper, we propose a new **task-agnostic** framework *SynBench* for benchmarking the robustness-accuracy performance of pretrained representations. SynBench is fundamentally task-independent and provides a quantifiable score that does not reply on any real-life data. SynBench exploits an idealized data distribution, class conditional Gaussian mixture, to establish a theoretically-derived robustness-accuracy tradeoff, which serves as the reference for pretrained representations. Finally, a quantifiable score *SynBench-Score* is provided that compares the ratio of area-under-curve between the reference and the pretrained representations. We validate the usefulness of SynBench on several pretrained vision transformers in giving insightful comparisons on different model attributes (e.g. model size, fine-tuned pretraining representations, ViT patch size, linear probing).

While we delved into the robustness-accuracy performance of pretrained representations of vision transformers, we envision the SynBench framework to be further extended to other trustworthiness dimensions such as privacy, fairness, etc. Moreover, as the popularization of pretrained representations in various domains (e.g. vision, language, speech), we foresee SynBench to be generalized to more domains, and shed light on task-agnostic benchmarking designs.

## Acknowledgement

Ching-Yun Ko would like to thank IBM Research and the summer internship program. This work was partially supported by the MIT-IBM Watson AI Lab and by the National Science Foundation.

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

# A   Related Work

**Pretrained models in vision.** In the past few years, much focus in the machine learning community has been shift to train representation networks capable of extracting features for a variety of downstream tasks with minimal fine-tuning. Nowadays, many common vision tasks are achieved with the assistant of good backbones, e.g. classifications [14, 15, 16, 17, 9, 18], object detection [19, 20], segmentation [21, 22], etc. Among the popular backbones, vision transformers (ViT) [9] have attracted enormous interest. ViTs stem from Transformers [23] and split an image into patches, which are then treated as tokens as for the original Transformers. We will exemplify the use of SynBench using several pretrained ViTs.

**Benchmarking pretrained models.** Since pretrained models are used as a foundation for different downstream tasks, it is central to transfer learning [24, 25], and also tightly related to model generalization [26, 27]. To benchmark the performance of a pretrained model, it is a convention to apply the pretrained model for a number of popular tasks and conduct linear probing on the representations [28, 9, 18, 10]. Besides linear probing, evaluation frameworks have been proposed based on mutual information [29] and minimum description length (MDL) [30, 31], which are reliant on the label information of the downstream tasks and are hence task-specific. Moreover, recent work [32] also discussed the sensitivity of validation accuracy (nonlinear probes) and MDL to evaluation dataset size, and proposed a variant of MDL and a sample complexity based quantifier that depends on the data distribution.

It was not until recently that more fundamental questions are brought up related to the pretrained models [1, 33, 34]. Lately, authors of [1] raised practical concerns about the homogenization incentivized by the scale of the pretraining. Although the homogenization might help in achieving competitive performance for some downstream tasks, the defects are also inherited by all these downstreams. On that account, a more careful study of the fundamentals of pretrained models is of paramount importance. Plex [33] was dedicated to explore the reliability of pretrained models by devising 10 types of tasks on 40 datasets in evaluating the desired aspect of reliability. Furthermore, it is pointed out [34] that pretrained models may not be robust to subpopulation or group shift as shown in 9 benchmarks. The adversarial robustness is benchmarked by authors of [35, 36], where [36] demonstrated the superior robustness of ViTs through Imagenet and [35] conducted white-box and transfer attacks on Imagenet and CIFAR10.

**Optimal representations.** In the seminal work of deep representation theory, [37] depicted the desired optimal representations in supervised learning to be sufficient for downstream task, invariant to the effect of nuisances, maximally disentangled, and has minimal mutual information between representations and inputs. Focusing more on generalization than compression, [38] gave the optimal representation based on $\mathcal{V}$-information [39] and probed generalization in deep learning. More recently, [40] defined the optimal representations for domain generalization. In [41], authors characterize the idealized representation properties for invariant self-supervised representation learning. Specifically, idealized representation should be well-distinguished by the desired family of probes for potential invariant tasks, have sufficiently large dimension, and be invariant to input augmentations. [42] discusses the necessary conditions for avoiding degenerate solutions when performing self-supervised pretraining.

SynBench differs from the above quantifiers as it does not need knowledge of any downstream data and has controls over the evaluation set size since we could draw arbitrary number of synthetic data. With the assumed synthetic data distribution, we could theoretically characterize the robustness-accuracy tradeoff that is independent to the downstream tasks. Therefore, SynBench provides a predefined standard of the tradeoff, which serves as the reference for representations induced by pretrained models. It should be also mentioned that, recently *sim-to-real* transfer paradigm has been leveraged to test the quality of real data, by projecting those onto the space of a model trained on large-scale synthetic data generated from a set of pre-defined grammar rules [43]. SynBench, though conceptually similar at a very high level, is different from that line of work – as the focus of this work is to quantify the accuracy-robustness tradeoff of pretrained representations using synthetic data from conditional distributions.

# B Proofs

**Result B.1.** *For samples $x$ following the conditional Gaussian in equation 1 with $\Sigma = I_d$ (d by d identity matrix) , given an $\ell_2$ adversarial budget $\epsilon \leq \|\mu\|_2$, the robust Bayes optimal classifier has the decision margin $\delta$ lower bounded by $\frac{|q/2 - x^T\mu(1-\epsilon/\|\mu\|_2)|}{(1-\epsilon/\|\mu\|_2)\|\mu\|_2}$, where $q = ln\{(1-p)/p\}$. With $p = \frac{1}{2}$, the lower bounds become $\frac{|x^T\mu|}{\|\mu\|_2}$.*

*Proof.* Consider the Bayes optimal $\ell_2$ $\epsilon$-robust classifier [8, Theorem 4.1]

$$\hat{y}^*(x) = sign\{x^T\mu(1 - \epsilon/\|\mu\|_2) - q/2\},$$

where $q = ln\{(1-p)/p\}$. For a realization $x$, we give the lower bound on the decision margin $\delta$

$$(x + \delta)^T\mu(1 - \epsilon/\|\mu\|_2) - q/2 = 0$$
$$\Leftrightarrow \quad \delta^T\mu(1 - \epsilon/\|\mu\|_2) = q/2 - x^T\mu(1 - \epsilon/\|\mu\|_2)$$
$$\Rightarrow \quad \|\delta\|_2 \geq \frac{|q/2 - x^T\mu(1 - \epsilon/\|\mu\|_2)|}{(1 - \epsilon/\|\mu\|_2)\|\mu\|_2}.$$

$\square$

**Result B.2.** *Assume a balanced dataset ($p = \frac{1}{2}, q = 0$) where samples $x$ follow the general conditional Gaussian $x|y = 1 \sim \mathcal{N}(\mu_1, \Sigma), x|y = -1 \sim \mathcal{N}(\mu_2, \Sigma)$, given an $\ell_2$ adversarial budget $\epsilon$, the robust Bayes optimal classifier gives a standard accuracy of $\Phi(\frac{\tilde{\mu}^T\Lambda^{-1}(\tilde{\mu}-z_\Lambda(\tilde{\mu}))}{\|\Lambda^{-1}(\tilde{\mu}-z_\Lambda(\tilde{\mu}))\|_\Lambda})$, where $\tilde{\mu} = F^T\frac{\mu_1-\mu_2}{2}$, $F\Lambda F^T = \Sigma$ is the economy-size (thin) decomposition with nonzero eigenvalues, and $z_\Lambda$ is the solution of the convex problem $\arg\min_{\|z\|_2 \leq \epsilon}(\tilde{\mu} - z)^T\Lambda^{-1}(\tilde{\mu} - z)$.*

*Proof.* With a general non-symmetric conditional Gaussians

$$x|y = 1 \sim \mathcal{N}(\mu_1, \Sigma), \quad x|y = -1 \sim \mathcal{N}(\mu_2, \Sigma),$$

we apply proper translation to symmetric conditional Gaussians

$$F^Tx|y = 1 \sim \mathcal{N}(F^T\mu_1, \Lambda), \quad F^Tx|y = -1 \sim \mathcal{N}(F^T\mu_2, \Lambda),$$

$$F^Tx - F^T\frac{\mu_1 + \mu_2}{2}|y = 1 \sim \mathcal{N}(\tilde{\mu}, \Lambda), \quad F^Tx - F^T\frac{\mu_1 + \mu_2}{2}|y = -1 \sim \mathcal{N}(-\tilde{\mu}, \Lambda),$$

where $\tilde{\mu} = F^T\frac{\mu_1-\mu_2}{2}$. Then, following [7, 44], we have the Bayes optimal robust classifier

$$\hat{y}^*(x) = sign\{(x - \frac{\mu_1 + \mu_2}{2})^TF\Lambda^{-1}(\tilde{\mu} - z_\Lambda(\tilde{\mu}))\}, \tag{4}$$

where $z_\Lambda$ is the solution of the convex problem $\arg\min_{\|z\|_2 \leq \epsilon}(\mu - z)^T\Lambda^{-1}(\mu - z)$. With this classifier, we can calculate the analytical standard accuracy by

$$\mathbb{P}(y = 1)\mathbb{P}[\hat{y}^*(x) = 1 \mid y = 1] + \mathbb{P}(y = -1)\mathbb{P}[\hat{y}^*(x) = -1 \mid y = -1]$$
$$= \mathbb{P}[\hat{y}^*(x) = 1 \mid y = 1]$$
$$= \mathbb{P}\left[(F^Tx - F^T\frac{\mu_1 + \mu_2}{2})^T\Lambda^{-1}(\tilde{\mu} - z_\Lambda(\tilde{\mu})) > 0 \mid y = 1\right]$$
$$= \mathbb{P}\left[(\tilde{\mu} + w)^T\Lambda^{-1}(\tilde{\mu} - z_\Lambda(\tilde{\mu})) > 0\right], \quad w \sim \mathcal{N}(0, \Lambda)$$
$$= \mathbb{P}\left[w^T\Lambda^{-1}(\tilde{\mu} - z_\Lambda(\tilde{\mu})) > -\tilde{\mu}^T\Lambda^{-1}(\tilde{\mu} - z_\Lambda(\tilde{\mu}))\right], \quad w \sim \mathcal{N}(0, \Lambda)$$
$$= \mathbb{P}\left[\frac{w^T\Lambda^{-1}(\tilde{\mu} - z_\Lambda(\tilde{\mu}))}{\|\Lambda^{-1}(\tilde{\mu} - z_\Lambda(\tilde{\mu}))\|_\Lambda} > -\frac{\tilde{\mu}^T\Lambda^{-1}(\tilde{\mu} - z_\Lambda(\tilde{\mu}))}{\|\Lambda^{-1}(\tilde{\mu} - z_\Lambda(\tilde{\mu}))\|_\Lambda}\right], \quad \frac{w^T\Lambda^{-1}(\tilde{\mu} - z_\Lambda(\tilde{\mu}))}{\|\Lambda^{-1}(\tilde{\mu} - z_\Lambda(\tilde{\mu}))\|_\Lambda} \sim \mathcal{N}(0, 1)$$
$$= \Phi(\frac{\tilde{\mu}^T\Lambda^{-1}(\tilde{\mu} - z_\Lambda(\tilde{\mu}))}{\|\Lambda^{-1}(\tilde{\mu} - z_\Lambda(\tilde{\mu}))\|_\Lambda})$$

$\square$

**Result B.3.** *Assume a balanced dataset ($p = \frac{1}{2}, q = 0$) where samples $x$ follow the general conditional Gaussian $x|y = 1 \sim \mathcal{N}(\mu_1, \Sigma), x|y = -1 \sim \mathcal{N}(\mu_2, \Sigma)$, given an $\ell_2$ adversarial budget $\epsilon$, the robust Bayes optimal classifier gives an expected scaled bound of $\mathbb{E}\left[\|\bar{\delta}_x\|_2 \mid \hat{y}^*(x) = y\right] \geq \frac{1}{\sqrt{2\pi}} \frac{1}{a\Phi^{-1}(a)} e^{-\frac{1}{2}\left(\Phi^{-1}(a)\right)^2} + 1$, where $a$ denotes the standard accuracy.*

*Proof.* Let $a$ denote the accuracy, $t$ denote $F^T x - F^T \frac{\mu_1 + \mu_2}{2}$, and $w$ denote $\Lambda^{-1}(\tilde{\mu} - z_\Lambda(\tilde{\mu}))$. From Result 2.2, we have that the standard accuracy for the Bayes optimal (robust) classifier is $\Phi(\frac{\tilde{\mu}^T w}{\|w\|_\Lambda})$, so $\frac{\sum^d \tilde{\mu}_i w_i}{\sqrt{\sum^d \lambda_i w_i^2}} = \Phi^{-1}(a)$. Since for binary classification, we only care about accuracy from 0.5 to 1, so we should have $\sum^d \tilde{\mu}_i w_i > 0$.

Now consider the classifier in equation 4, the corresponding lower bound and scaled lower bound can be given as

$$\|\delta_x\|_2 \geq \frac{|(x - \frac{\mu_1+\mu_2}{2})^T F \Lambda^{-1}(\tilde{\mu} - z_\Lambda(\tilde{\mu}))|}{\|F\Lambda^{-1}(\tilde{\mu} - z_\Lambda(\tilde{\mu}))\|_2},$$

$$\|\bar{\delta}_x\|_2 \geq \frac{|(x - \frac{\mu_1+\mu_2}{2})^T F \Lambda^{-1}(\tilde{\mu} - z_\Lambda(\tilde{\mu}))|}{\|F\Lambda^{-1}(\tilde{\mu} - z_\Lambda(\tilde{\mu}))\|_2} \frac{\|F\Lambda^{-1}(\tilde{\mu} - z_\Lambda(\tilde{\mu}))\|_2}{|\tilde{\mu}^T \Lambda^{-1}(\tilde{\mu} - z_\Lambda(\tilde{\mu}))|}$$

$$= \frac{|(x - \frac{\mu_1+\mu_2}{2})^T F \Lambda^{-1}(\tilde{\mu} - z_\Lambda(\tilde{\mu}))|}{|\tilde{\mu}^T \Lambda^{-1}(\tilde{\mu} - z_\Lambda(\tilde{\mu}))|}.$$

When $t = F^T x - F^T \frac{\mu_1 + \mu_2}{2}$, and $w = \Lambda^{-1}(\tilde{\mu} - z_\Lambda(\tilde{\mu}))$,

$$\|\bar{\delta}_x\|_2 \geq \frac{|t^T w|}{|\tilde{\mu}^T w|} = \frac{|\sum^d t_i w_i|}{|\sum^d \tilde{\mu}_i w_i|} = \frac{|\sum^d t_i w_i|}{\sum^d \tilde{\mu}_i w_i}.$$

Since $t|y \sim \mathcal{N}(y\tilde{\mu}, \Lambda)$, we have $t_i w_i|y \sim \mathcal{N}(y\tilde{\mu}_i w_i, \lambda_i w_i^2)$ and

$$\sum^d t_i w_i|y \sim \mathcal{N}(\sum^d y\tilde{\mu}_i w_i, \sum^d \lambda_i w_i^2).$$

When we only want to get the expected scaled bound of the correctly-classified samples, we have that

$$\mathbb{E}\left[\|\bar{\delta}\|_2 \mid \hat{y}^*(x) = y\right] \geq \frac{1}{\sum^d \tilde{\mu}_i w_i} \mathbb{E}\left[|\sum^d t_i w_i| \mid \hat{y}^*(x) = y\right]$$

$$= \frac{p}{\sum^d \tilde{\mu}_i w_i} \mathbb{E}\left[|\sum^d t_i w_i| \mid \hat{y}^*(x) = y = 1\right]$$

$$+ \frac{1 - p}{\sum^d \tilde{\mu}_i w_i} \mathbb{E}\left[|\sum^d t_i w_i| \mid \hat{y}^*(x) = y = -1\right]$$

$$= \frac{p}{\sum^d \tilde{\mu}_i w_i} \mathbb{E}\left[\sum^d t_i w_i \mid y = 1, \sum^d t_i w_i \geq 0\right]$$

$$+ \frac{1 - p}{\sum^d \tilde{\mu}_i w_i} \mathbb{E}\left[-\sum^d t_i w_i \mid y = -1, \sum^d t_i w_i < 0\right].$$

Recall that $\sum^d t_i w_i|y \sim \mathcal{N}(\sum^d y\tilde{\mu}_i w_i, \sum^d \lambda_i w_i^2)$, then by the mean of truncated normal distribution, it is true that

$$\mathbb{E}\left[\sum^d t_i w_i \mid y = 1, \sum^d t_i w_i \geq 0\right] = \sum^d \tilde{\mu}_i w_i + \sqrt{\sum^d \lambda_i w_i^2} \frac{\phi(\frac{0 - \sum^d \tilde{\mu}_i w_i}{\sqrt{\sum^d \lambda_i w_i^2}})}{1 - \Phi(\frac{0 - \sum^d \tilde{\mu}_i w_i}{\sqrt{\sum^d \lambda_i w_i^2}})}$$

$$= \sum^d \tilde{\mu}_i w_i + \sqrt{\sum^d \lambda_i w_i^2} \frac{\phi(-\frac{\sum^d \tilde{\mu}_i w_i}{\sqrt{\sum^d \lambda_i w_i^2}})}{1 - \Phi(-\frac{\sum^d \tilde{\mu}_i w_i}{\sqrt{\sum^d \lambda_i w_i^2}})}$$

$$= \sum^{d} \tilde{\mu}_i w_i$$

$$+ \sqrt{\sum^{d} \lambda_i w_i^2} \frac{1}{\sqrt{2\pi}\Phi\left(\frac{\sum^{d} \tilde{\mu}_i w_i}{\sqrt{\sum^{d} \lambda_i w_i^2}}\right)} e^{-\frac{1}{2}\left(\frac{\sum^{d} \tilde{\mu}_i w_i}{\sqrt{\sum^{d} \lambda_i w_i^2}}\right)^2}$$

$$\mathbb{E}\left[-\sum^{d} t_i w_i \mid y=-1, \sum^{d} t_i w_i < 0\right] = -\mathbb{E}\left[\sum^{d} t_i w_i \mid y=-1, \sum^{d} t_i w_i < 0\right]$$

$$= -\left(-\sum^{d} \tilde{\mu}_i w_i - \sqrt{\sum^{d} \lambda_i w_i^2} \frac{\phi\left(\frac{0+\sum^{d} \tilde{\mu}_i w_i}{\sqrt{\sum^{d} \lambda_i w_i^2}}\right)}{\Phi\left(\frac{0+\sum^{d} \tilde{\mu}_i w_i}{\sqrt{\sum^{d} \lambda_i w_i^2}}\right)}\right)$$

$$= \sum^{d} \tilde{\mu}_i w_i$$

$$+ \sqrt{\sum^{d} \lambda_i w_i^2} \frac{1}{\sqrt{2\pi}\Phi\left(\frac{\sum^{d} \tilde{\mu}_i w_i}{\sqrt{\sum^{d} \lambda_i w_i^2}}\right)} e^{-\frac{1}{2}\left(\frac{\sum^{d} \tilde{\mu}_i w_i}{\sqrt{\sum^{d} \lambda_i w_i^2}}\right)^2}.$$

Therefore

$$\mathbb{E}\left[\|\bar{\delta}_x\|_2 \mid \hat{y}^*(x)=y\right] \geq \frac{1}{\sum^{d} \tilde{\mu}_i w_i}\left(\sum^{d} \tilde{\mu}_i w_i + \sqrt{\sum^{d} \lambda_i w_i^2} \frac{1}{\sqrt{2\pi}\Phi\left(\frac{\sum^{d} \tilde{\mu}_i w_i}{\sqrt{\sum^{d} \lambda_i w_i^2}}\right)} e^{-\frac{1}{2}\left(\frac{\sum^{d} \tilde{\mu}_i w_i}{\sqrt{\sum^{d} \lambda_i w_i^2}}\right)^2}\right)$$

$$= 1 + \frac{\sqrt{\sum^{d} \lambda_i w_i^2}}{\sum^{d} \tilde{\mu}_i w_i} \frac{1}{\sqrt{2\pi}\Phi\left(\frac{\sum^{d} \tilde{\mu}_i w_i}{\sqrt{\sum^{d} \lambda_i w_i^2}}\right)} e^{-\frac{1}{2}\left(\frac{\sum^{d} \tilde{\mu}_i w_i}{\sqrt{\sum^{d} \lambda_i w_i^2}}\right)^2}.$$

By replacing $\frac{\sum^{d} \tilde{\mu}_i w_i}{\sqrt{\sum^{d} \lambda_i w_i^2}}$ by $\Phi^{-1}(a)$, we got

$$\mathbb{E}\left[\|\bar{\delta}_x\|_2 \mid \hat{y}^*(x)=y\right] \geq \frac{1}{\sqrt{2\pi}} \frac{1}{a\Phi^{-1}(a)} e^{-\frac{1}{2}\left(\Phi^{-1}(a)\right)^2} + 1.$$

$\square$

**Result B.4.** *For representations $z$ following the general class-balanced conditional Gaussian $z|y = 1 \sim \mathcal{N}(\mu_1, \Sigma)$, $z|y = -1 \sim \mathcal{N}(\mu_2, \Sigma)$, given an $\ell_2$ adversarial budget $\epsilon$, the robust Bayes optimal classifier has the decision margin $\delta_z$ lower bounded by $\frac{|(z-\frac{\mu_1+\mu_2}{2})^T F\Lambda^{-1}(\tilde{\mu}-z_\Lambda(\tilde{\mu}))|}{\|F\Lambda^{-1}(\tilde{\mu}-z_\Lambda(\tilde{\mu}))\|_2}$, and a scaled bound of $\|\bar{\delta}_z\|_2 \geq \frac{|(z-\frac{\mu_1+\mu_2}{2})^T F\Lambda^{-1}(\tilde{\mu}-z_\Lambda(\tilde{\mu}))|}{|\tilde{\mu}^T \Lambda^{-1}(\tilde{\mu}-z_\Lambda(\tilde{\mu}))|}$, where $\tilde{\mu} = F^T \frac{\mu_1-\mu_2}{2}$, $F\Lambda F^T = \Sigma$ is the economy-size (thin) decomposition with nonzero eigenvalues, and $z_\Lambda$ is the solution of the convex problem $\arg\min_{\|z\|_2 \leq \epsilon}(\tilde{\mu}-z)^T \Lambda^{-1}(\tilde{\mu}-z)$.*

*Proof.* The proof follows similarly as before and is the intermediate result in the proof of Result 2.3. With a general non-symmetric conditional Gaussians

$$z|y = 1 \sim \mathcal{N}(\mu_1, \Sigma), \ z|y = -1 \sim \mathcal{N}(\mu_2, \Sigma),$$

and after translation

$$F^T z - F^T \frac{\mu_1 + \mu_2}{2}|y = 1 \sim \mathcal{N}(\tilde{\mu}, \Lambda), \ F^T z - F^T \frac{\mu_1 + \mu_2}{2}|y = -1 \sim \mathcal{N}(-\tilde{\mu}, \Lambda),$$

where $\tilde{\mu} = F^T \frac{\mu_1-\mu_2}{2}$. Following [7, 44], we have the Bayes optimal robust classifier

$$\hat{y}^*(z) = sign\{(z - \frac{\mu_1 + \mu_2}{2})^T F\Lambda^{-1}(\tilde{\mu} - z_\Lambda(\tilde{\mu}))\}, \tag{5}$$

where $z_\Lambda$ is the solution of the convex problem $\arg\min_{\|z\|_2 \leq \epsilon}(\mu - z)^T\Lambda^{-1}(\mu - z)$. The corresponding lower bounds is

$$\|\delta_z\|_2 \geq \frac{|(z - \frac{\mu_1 + \mu_2}{2})^T F\Lambda^{-1}(\tilde{\mu} - z_\Lambda(\tilde{\mu}))|}{\|F\Lambda^{-1}(\tilde{\mu} - z_\Lambda(\tilde{\mu}))\|_2},$$

$$\|\bar{\delta}_z\|_2 \geq \frac{|(z - \frac{\mu_1 + \mu_2}{2})^T F\Lambda^{-1}(\tilde{\mu} - z_\Lambda(\tilde{\mu}))|}{\|F\Lambda^{-1}(\tilde{\mu} - z_\Lambda(\tilde{\mu}))\|_2} \frac{\|F\Lambda^{-1}(\tilde{\mu} - z_\Lambda(\tilde{\mu}))\|_2}{|\tilde{\mu}^T\Lambda^{-1}(\tilde{\mu} - z_\Lambda(\tilde{\mu}))|}$$

$$= \frac{|(z - \frac{\mu_1 + \mu_2}{2})^T F\Lambda^{-1}(\tilde{\mu} - z_\Lambda(\tilde{\mu}))|}{|\tilde{\mu}^T\Lambda^{-1}(\tilde{\mu} - z_\Lambda(\tilde{\mu}))|}$$

$\square$

## C $\quad \ell_p$ Results

We note that our results can be straightforwardly generalized to $\ell_p$.

**Result C.1.** *Assume a balanced dataset where samples $x$ follow the general conditional Gaussian $x|y = 1 \sim \mathcal{N}(\mu_1, \Sigma), x|y = -1 \sim \mathcal{N}(\mu_2, \Sigma)$, given an $\ell_p$ adversarial budget $\epsilon$, the robust Bayes optimal classifier gives a standard accuracy of $\Phi(\frac{\tilde{\mu}^T\Lambda^{-1}(\tilde{\mu} - z_\Lambda(\tilde{\mu}))}{\|\Lambda^{-1}(\tilde{\mu} - z_\Lambda(\tilde{\mu}))\|_\Lambda})$, where $\tilde{\mu} = F^T\frac{\mu_1 - \mu_2}{2}$, $F\Lambda F^T = \Sigma$ is the economy-size (thin) decomposition with nonzero eigenvalues, and $z_\Lambda$ is the solution of the convex problem $\arg\min_{\|z\|_p \leq \epsilon}(\tilde{\mu} - z)^T\Lambda^{-1}(\tilde{\mu} - z)$.*

**Result C.2.** *Assume a balanced dataset where samples $x$ follow the general conditional Gaussian $x|y = 1 \sim \mathcal{N}(\mu_1, \Sigma), x|y = -1 \sim \mathcal{N}(\mu_2, \Sigma)$, given an $\ell_p$ adversarial budget $\epsilon$, the robust Bayes optimal classifier gives an expected scaled bound of $\mathbb{E}\left[\|\bar{\delta}_x\|_p \mid \hat{y}^*(x) = y\right] \geq \frac{1}{\sqrt{2\pi}}\frac{1}{a\Phi^{-1}(a)}e^{-\frac{1}{2}\left(\Phi^{-1}(a)\right)^2} + 1$, where $a$ denotes the standard accuracy.*

*Proof.* We follow the proof of Thm B.3 and observe that we now have the corresponding lower bound and scaled lower bound can be given as

$$\|\delta_x\|_p \geq \frac{|(x - \frac{\mu_1 + \mu_2}{2})^T F\Lambda^{-1}(\tilde{\mu} - z_\Lambda(\tilde{\mu}))|}{\|F\Lambda^{-1}(\tilde{\mu} - z_\Lambda(\tilde{\mu}))\|_q},$$

$$\|\bar{\delta}_x\|_p \geq \frac{|(x - \frac{\mu_1 + \mu_2}{2})^T F\Lambda^{-1}(\tilde{\mu} - z_\Lambda(\tilde{\mu}))|}{\|F\Lambda^{-1}(\tilde{\mu} - z_\Lambda(\tilde{\mu}))\|_q} \frac{\|F\Lambda^{-1}(\tilde{\mu} - z_\Lambda(\tilde{\mu}))\|_q}{|\tilde{\mu}^T\Lambda^{-1}(\tilde{\mu} - z_\Lambda(\tilde{\mu}))|}$$

$$= \frac{|(x - \frac{\mu_1 + \mu_2}{2})^T F\Lambda^{-1}(\tilde{\mu} - z_\Lambda(\tilde{\mu}))|}{|\tilde{\mu}^T\Lambda^{-1}(\tilde{\mu} - z_\Lambda(\tilde{\mu}))|},$$

where $\frac{1}{p} + \frac{1}{q} = 1$. The remainder of the proof will then follows as in Thm B.3 which renders

$$\mathbb{E}\left[\|\bar{\delta}_x\|_p \mid \hat{y}^*(x) = y\right] \geq \frac{1}{\sqrt{2\pi}}\frac{1}{a\Phi^{-1}(a)}e^{-\frac{1}{2}\left(\Phi^{-1}(a)\right)^2} + 1.$$

$\square$

**Result C.3.** *For representations $z$ following the general class-balanced conditional Gaussian $z|y = 1 \sim \mathcal{N}(\mu_1, \Sigma), \ z|y = -1 \sim \mathcal{N}(\mu_2, \Sigma)$, given an $\ell_p$ adversarial budget $\epsilon$, the robust Bayes optimal classifier has the decision margin $\delta_z$ lower bounded by $\frac{|(z - \frac{\mu_1 + \mu_2}{2})^T F\Lambda^{-1}(\tilde{\mu} - z_\Lambda(\tilde{\mu}))|}{\|F\Lambda^{-1}(\tilde{\mu} - z_\Lambda(\tilde{\mu}))\|_q}$, and a scaled bound of $\|\bar{\delta}_z\|_p \geq \frac{|(z - \frac{\mu_1 + \mu_2}{2})^T F\Lambda^{-1}(\tilde{\mu} - z_\Lambda(\tilde{\mu}))|}{|\tilde{\mu}^T\Lambda^{-1}(\tilde{\mu} - z_\Lambda(\tilde{\mu}))|}$, where $\frac{1}{p} + \frac{1}{q} = 1$, $\tilde{\mu} = F^T\frac{\mu_1 - \mu_2}{2}$, $F\Lambda F^T = \Sigma$ is the economy-size (thin) decomposition with nonzero eigenvalues, and $z_\Lambda$ is the solution of the convex problem $\arg\min_{\|z\|_p \leq \epsilon}(\tilde{\mu} - z)^T\Lambda^{-1}(\tilde{\mu} - z)$.*

## D   Robust linear probing procedure

For a given pretrained model, let $f$ and $g$ be the pretrained network and linear probing layer, we solve the optimization problem $\min_g \max_{\|\delta\| \leq \epsilon} L(g(f(x + \delta)), y)$ using the PyTorch library Torchattacks[2] and 10-step PGDL2 attacks [45] for adversarial training.

---

[2]https://github.com/Harry24k/adversarial-attacks-pytorch

# E   Tables

| Model | Arch. | pretraining | fine-tuning | patch | # parameters (M) |
|-------|-------|-------------|-------------|-------|------------------|
| ViT-Ti/16 | ViT-Tiny | Imgn21k | Imgn1k | 16 | 5.7 |
| ViT-B/16 | ViT-Base | Imgn21k | Imgn1k | 16 | 86.6 |
| ViT-B/16-in21k | ViT-Base | Imgn21k | No | 16 | 86.6 |
| ViT-B/32 | ViT-Base | Imgn21k | Imgn1k | 32 | 88.2 |
| ViT-L/16 | ViT-Large | Imgn21k | Imgn1k | 16 | 304.3 |
| Variation: | | | | | |
| Model size | (ViT-Ti/16, ViT-Base/16, ViT-Large/16) | | | | |
| Finetuning | (ViT-B/16, ViT-B/16-in21k) | | | | |
| ViT patch size | (ViT-B/16, ViT-B/32) | | | | |

Table 4: Model descriptions.

| $a_t$ | Model | $\epsilon = 0$ | $\epsilon = 0.2$ | $\epsilon = 0.4$ | $\epsilon = 0.6$ | $\epsilon = 0.8$ | CIFAR10 | CIFAR10-c |
|-------|-------|------|--------|--------|--------|--------|---------|-----------|
| 0.7 | ViT-B/16 | 0.33 | 0.37 | 0.32 | 0.20 | 0.06 | 95.0 | 81.2 |
| | ViT-B/16-in21k | 0.20 | 0.23 | 0.18 | 0.07 | 0.01 | 89.6 | 71.4 |
| 0.75 | ViT-B/16 | 0.26 | 0.30 | 0.25 | 0.11 | 0.01 | 95.0 | 81.2 |
| | ViT-B/16-in21k | 0.12 | 0.16 | 0.10 | 0.02 | 0 | 89.6 | 71.4 |
| 0.8 | ViT-B/16 | 0.19 | 0.23 | 0.17 | 0.04 | 0 | 95.0 | 81.2 |
| | ViT-B/16-in21k | 0.06 | 0.09 | 0.04 | 0 | 0 | 89.6 | 71.4 |
| 0.85 | ViT-B/16 | 0.10 | 0.15 | 0.09 | 0 | 0 | 95.0 | 81.2 |
| | ViT-B/16-in21k | 0.01 | 0.02 | 0 | 0 | 0 | 89.6 | 71.4 |
| 0.9 | ViT-B/16 | 0.02 | 0.04 | 0.01 | 0 | 0 | 95.0 | 81.2 |
| | ViT-B/16-in21k | 0 | 0 | 0 | 0 | 0 | 89.6 | 71.4 |

Table 5: Full table of Table 1.

| $a_t$ | Model | $\epsilon = 0$ | $\epsilon = 0.2$ | $\epsilon = 0.4$ | $\epsilon = 0.6$ | $\epsilon = 0.8$ | CIFAR10 | CIFAR10-c |
|-------|-------|------|--------|--------|--------|--------|---------|-----------|
| 0.7 | ViT-Ti/16 | 0.01 | 0 | 0 | 0 | 0 | 81.9 | 59.1 |
| | ViT-B/16 | 0.33 | 0.37 | 0.32 | 0.20 | 0.06 | 95.0 | 81.2 |
| | ViT-L/16 | 0.26 | 0.33 | 0.30 | 0.22 | 0.11 | 98.0 | 90.3 |
| 0.75 | ViT-Ti/16 | 0 | 0 | 0 | 0 | 0 | 81.9 | 59.1 |
| | ViT-B/16 | 0.26 | 0.30 | 0.25 | 0.11 | 0.01 | 95.0 | 81.2 |
| | ViT-L/16 | 0.19 | 0.27 | 0.24 | 0.16 | 0.04 | 98.0 | 90.3 |
| 0.8 | ViT-Ti/16 | 0 | 0 | 0 | 0 | 0 | 81.9 | 59.1 |
| | ViT-B/16 | 0.19 | 0.23 | 0.17 | 0.04 | 0 | 95.0 | 81.2 |
| | ViT-L/16 | 0.12 | 0.21 | 0.18 | 0.09 | 0 | 98.0 | 90.3 |
| 0.85 | ViT-Ti/16 | 0 | 0 | 0 | 0 | 0 | 81.9 | 59.1 |
| | ViT-B/16 | 0.10 | 0.15 | 0.09 | 0 | 0 | 95.0 | 81.2 |
| | ViT-L/16 | 0.05 | 0.13 | 0.10 | 0.03 | 0 | 98.0 | 90.3 |
| 0.9 | ViT-Ti/16 | 0 | 0 | 0 | 0 | 0 | 81.9 | 59.1 |
| | ViT-B/16 | 0.02 | 0.04 | 0.01 | 0 | 0 | 95.0 | 81.2 |
| | ViT-L/16 | 0 | 0.04 | 0.03 | 0 | 0 | 98.0 | 90.3 |

Table 6: Full table of Table 2

| | arg max$_x$ SynBench-Score | linear probing | | 0.1-robust linear probing | | 0.2-robust linear probing | | 0.3-robust linear probing | | 0.4-robust linear probing | |
|---|---|---|---|---|---|---|---|---|---|---|---|
| | | CIFAR10 | CIFAR10-c | CIFAR10 | CIFAR10-c | CIFAR10 | CIFAR10-c | CIFAR10 | CIFAR10-c | CIFAR10 | CIFAR10-c |
| ViT-Ti/16 | 0.1 | 80.5±0.17 | 58.9±0.39 | 79.0±0.13 | 59.1±0.22 | 76.2±0.13 | 58.2±0.41 | - | - | - | - |
| ViT-B/16 | 0.2 | 94.5±0.05 | 80.7±0.15 | 95.2±0.04 | 81.6±0.33 | 95.1±0.05 | 82.0±0.18 | 95.0±0.05 | 81.9±0.43 | - | - |
| ViT-L/16 | 0.2 | 98.0±0.03 | 90.3±0.14 | 98.2±0.02 | 90.4±0.56 | 98.4±0.04 | 90.7±0.26 | 98.3±0.10 | 90.9±0.18 | - | - |
| ViT-B/16-in21k | 0.1/0.2 | 88.9±0.36 | 70.8±0.37 | 89.3±0.03 | 71.5±0.43 | 89.1±0.14 | 71.6±0.29 | 89.2±0.16 | 71.4±0.10 | - | - |
| ViT-B/32 | 0.2/0.3 | 92.0±0.15 | 75.8±0.33 | 92.5±0.04 | 76.9±0.30 | 92.4±0.06 | 76.9±0.27 | 92.4±0.07 | 77.1±0.18 | 92.4±0.05 | 77.0±0.05 |

Table 7: Full table of Table 3

| $a_t = 0.6$ | $\epsilon = 0$ | $\epsilon = 0.2$ | $\epsilon = 0.4$ | $\epsilon = 0.6$ | $\epsilon = 0.8$ | CIFAR10 | CIFAR10-c |
|---|---|---|---|---|---|---|---|
| ViT-B/16 | 0.45 | 0.47 | 0.44 | 0.36 | 0.25 | 95.0 | 81.2 |
| ViT-B/32 | 0.02 | 0.03 | 0.03 | 0.01 | 0 | 92.2 | 76.6 |

Table 8: Comparisons on the ViT patch size. The SynBench-Score of ViTs of different patch sizes, and the standard linear probing accuracy on CIFAR10 and transfer accuracy on CIFAR10-c. The Imagenet result shows an average accuracy over 6 Imagenet variants.

| | attack success rate on standard linear probing | attack success rate on robust linear probing |
|---|---|---|
| ViT-Ti/16 | 98.9 | 98.9 |
| ViT-B/16 | 80.1 | 60.1 |
| ViT-L/16 | 53.2 | 37.2 |
| ViT-B/16-in21k | 92.1 | 81.5 |
| ViT-B/32 | 58.5 | 44.3 |

Table 9: Auto-attack success rates on standard CIFAR10 classifiers that built on pretrained models.

