# OpenReview forum: "SynBench: Task-Agnostic Benchmarking of Pretrained Representations using Synthetic Data"
_NeurIPS.cc/2022/Workshop/SyntheticData4ML — Neurips 2022 SyntheticData4ML_

### Official Review · Reviewer_oCAD · 2022-10-18
**The clarity of this paper should be improved**

**Rating:** 5
**Confidence:** 2

**Review:**

The paper proposed a novel way to evaluate the pretrained representations using synthetic data. The method calculates the difference between the bound-threshold accuracy plot for the two sets of input. The theoretical results are provided. The reviewer believes the clarity of this paper should be improved. Especially,

1. The intuition of the score is missing. Why this score is important and how is the score related to the probing performance of the pretraining language model?

2. Need to state the way to calculate the score clearly in a numerical way. From the paper, the reviewer finds it difficult to reproduce the score.

3. The author(s) show the scores for different ViT models. But it would be good to show those scores can directly link to the probing performance of those models.

---

### Official Review · Reviewer_w98S · 2022-10-18
**Interesting theoretical result but I'm not sure if applicable in practice**

**Rating:** 5
**Confidence:** 3

**Review:**

- The paper presents a benchmarking approach for comparing pre-trained models to each other. The authors propose to use a mixture of Gaussian distributions and a set of optimal linear classifiers to gauge the quality of representations coming from these models.
- Authors postulate that since in many cases linear probing is used to fine-tune said models to downstream tasks, such problem formulation is sufficient to establish a functional benchmark. Authors experiment with a variety of pre-trained models and validate their results by comparing to actual performance on a CIFAR10 task.
- Overall, it is a well-written paper, but it does seem to be limited to cases where linear probing is used on top of the pre-trained model.
- Additionally, while it might work in vision use cases, the assumption that minor perturbation shouldn’t change the label may not be reasonable for all tasks and could depend on how you place the Gaussians.
- Even for vision use cases it seems that Gaussians is just not a good way to explore the input space. In Gowal et al.[1], Table 1 seems to suggest that the Gaussian fit has very poor coverage and complementarity compared to any other generative model.
- Finally, it does seem like the SynBench score correlates poorly with the CIFAR scores (Table 2).
- Thus, while the theoretical framework is interesting and I enjoyed reading and diving deep into the paper – I’m not convinced there is a practical benefit in using SynBench. Additionally, I think this paper is way too short for the topic (e.g., you really have to go into Appendices to get the picture!) it tries to cover – testing on more datasets, going beyond the Gaussians and computer vision tasks could strengthen the case for SynBench a lot, although authors did a great job of condensing the paper from it's (apparent) original size without loss of coherence. But it still results in the paper that is hard to read and pretty much requires a motivated reader with some expertise in recent adversarial training literature for computer vision.
- [1] https://arxiv.org/pdf/2110.09468.pdf

---

### Official Review · Reviewer_uWAu · 2022-10-18
**Benchmarking the use of synthetic samples**

**Rating:** 7
**Confidence:** 1

**Review:**

Authors have presented a nice benchmarking scheme based on robustness measure to derive the goodness of the synthetic data used for training large models. Rigorous derivation is provided with the method being backed strongly.

---

### Meta-Review · Area_Chair_6vkZ · 2022-10-20

**Recommendation:** Accept